# Targeting the Gastrin-Releasing Peptide Receptor (GRP-R) in Cancer Therapy: Development of Bombesin-Based Peptide–Drug Conjugates

**DOI:** 10.3390/ijms24043400

**Published:** 2023-02-08

**Authors:** Jacopo Gomena, Balázs Vári, Rita Oláh-Szabó, Beáta Biri-Kovács, Szilvia Bősze, Adina Borbély, Ádám Soós, Ivan Ranđelović, József Tóvári, Gábor Mező

**Affiliations:** 1Institute of Chemistry, Faculty of Science, Eötvös Loránd University, 1117 Budapest, Hungary; 2ELKH-ELTE Research Group of Peptide Chemistry, 1117 Budapest, Hungary; 3Department of Experimental Pharmacology, National Institute of Oncology, 1122 Budapest, Hungary; 4Department of Genetics, Cell and Immunobiology, Semmelweis University, 1089 Budapest, Hungary; 5MTA-ELTE Lendület Ion Mobility Mass Spectrometry Research Group, 1117 Budapest, Hungary; 6Department of Anatomy, Histology and Embryology, Semmelweis University, 1085 Budapest, Hungary; 7KINETO Lab Ltd., 1037 Budapest, Hungary

**Keywords:** bombesin, gastrin-releasing peptide receptor, targeted tumour therapy, peptide–drug conjugates, prostate cancer, breast cancer, drug delivery systems

## Abstract

Targeted tumour therapy has proved to be an efficient alternative to overcome the limitations of conventional chemotherapy. Among several receptors upregulated in cancer cells, the gastrin-releasing peptide receptor (GRP-R) has recently emerged as a promising target for cancer imaging, diagnosing and treatment due to its overexpression on cancerous tissues such as breast, prostate, pancreatic and small-cell lung cancer. Herein, we report on the in vitro and in vivo selective delivery of the cytotoxic drug daunorubicin to prostate and breast cancer, by targeting GRP-R. Exploiting many bombesin analogues as homing peptides, including a newly developed peptide, we produced eleven daunorubicin-containing peptide–drug conjugates (PDCs), acting as drug delivery systems to safely reach the tumour environment. Two of our bioconjugates revealed remarkable anti-proliferative activity, an efficient uptake by all three tested human breast and prostate cancer cell lines, high stability in plasma and a prompt release of the drug-containing metabolite by lysosomal enzymes. Moreover, they revealed a safe profile and a consistent reduction of the tumour volume in vivo. In conclusion, we highlight the importance of GRP-R binding PDCs in targeted cancer therapy, with the possibility of further tailoring and optimisation.

## 1. Introduction

Cancer is among the leading causes of death worldwide. According to estimates from the International Agency for Research on Cancer, in 2020, more than 19 million cases and nearly 10 million deaths were reported. Among those, prostate cancer ranked second in terms of incidence, and fifth in terms of deaths caused by cancer globally among men. Moreover, in 112 out of 185 countries, it is the most frequently diagnosed tumour in men. On the other hand, female breast cancer became the leading cause of cancer incidence, surpassing lung cancer, with 11.7% of all cancer cases, and represents the fifth globally leading cause of cancer mortality [1].

Chemotherapy has always been the most common treatment for cancer. However, the nonspecific distribution of many chemotherapeutics limits their clinical applications and causes high levels of toxicity. Therefore, targeted tumour therapy has appeared as a promising approach to overcome such limitations. Drug delivery systems (DDS) exploit differences between healthy and cancerous cells and tissues, to selectively deliver a toxic payload to the site of action and attempt to minimise off-target side effects. As a result, many antibody–drug conjugates (ADCs) have already obtained FDA and/or EMA approval, such as trastuzumab emtansine (Kadcyla^®^), brentuximab vedotin (Adcetris^®^) and, most recently, loncastuximab tesirine (Zylonta^®^), disitamab vedotin (Aidixi^®^) and tisotumab vedotin (Tivdak^®^) [2,3,4]. Similarly to the emerging concept of ADC technology, peptide–drug conjugates (PDCs) have gained increasing interest due to the distinct benefits of tumour homing peptides. As a matter of fact, having a peptide ligand instead of a monoclonal antibody (mAb) allows us to overcome some of the limitations of ADCs in cancer therapy, such as high production costs, poor tumour permeability and potentially dangerous immune reactions [5,6,7,8]. These advantages recently led to the authorisation of three PDCs for clinical use in cancer: vipivotide tetraxetan, melphalan flufenamide and a somatostatin derivative, ^177^Lu-DOTATATE [9,10,11,12]. Several others, such as Bicycle Therapeutics’ BT1718, BT5528 and BT8009, Cybrexa’s CBX-12 and Shenogen’s SNG1005, are undergoing clinical trials [13,14].

Choosing an appropriate target is crucial for the development of effective and safe devices for the delivery of chemotherapeutics. For example, tumour cells can be discriminated from normal cells due to upregulated cell surface receptors or enzyme levels. After obtaining encouraging results with GnRH-III-based PDCs directed towards the gonadotropin-releasing hormone receptor (GnRH-R) [15,16,17,18], our research group has focused on the gastrin-releasing peptide receptor (GRP-R, or bombesin receptor 2, BB2), which is overexpressed in several malignancies, such as prostate, breast and lung cancer, while being poorly expressed physiologically in healthy tissues [8,19,20,21]. This receptor is part of the bombesin receptor protein family, together with the neuromedin B receptor (NMB-R, BB1) and the bombesin receptor subtype 3 (BRS-3, BB3). Its native ligand is the gastrin-releasing peptide, of which the C-terminal heptapeptide fragment is common with bombesin (BBN), a 14-mer peptide which was first discovered in the skin of the European fire-bellied toad, *Bombina bombina* [20,22]. The mentioned truncated version, BBN (7-14) (Gln-Trp-Ala-Val-Gly-His-Leu-Met-NH_2_), maintains the affinity towards GRP-R. Hence, it has been studied as a targeting ligand and has been the starting point for the synthesis of several analogues throughout the years [21,23,24,25,26,27,28,29,30,31,32,33,34,35,36,37], with retained or improved affinity for GRP-R [21,26,29,30,31,32,33,34,35,37] and increased stability in plasma [27,29,31,32,33,34,36,37], but also with modulated agonist or antagonist activity [21,26,28,32,34,35,36]. For example, a very common modification is the insertion of a D-Phe in position 6, reported to increase the affinity [21]. As for position 11, Hoppenz et al. have reported that the presence of β-Ala, together with an Ala in position 13, provides peptides that are highly selective for GRP-R, whereas the selectivity is lost with either Leu^13^ or Phe^13^ and the binding is lost with D-Ala^13^. Moreover, the exchange of Ala^13^ to either Aib or *N*-Me–Ala, conferred longer stability in plasma [27]. The exchange of Leu^13^–Met^14^ to Sta^13^–Leu^14^ represents another successful substitution: the affinity towards GRP-R is increased and Sta^13^ improves the resistance to the neutral endopeptidase-driven cleavage of the His^12^–Leu^13^ bond. Furthermore, together with D-Phe^6^, it confers an antagonistic activity to the bombesin analogues [31]. Finally, the Gly^11^–His^12^ bond was reinforced by substituting Gly^11^ with *N*-Me–Gly [31,38]. Such analogues have been developed and tested for cancer imaging, diagnosis [39,40] and treatment [23,30,41,42,43,44,45,46,47].

We decided to select some of the described peptides [27,28,31,40,41] bearing the mentioned substitutions and having affirmed affinity for GRP-R in the low nanomolar range, together with the original BBN (7-14), and use them as targeting moiety to deliver a cytotoxic payload, daunorubicin (Dau), to prostate and breast cancers. Compared to the BBN (7-14) sequence, our targeting peptides are elongated by a D-Phe in position 6 and comprise substitutions in positions 11, 13 and 14 (Figure 1). Starting from these variations, we have also generated a new sequence: [D-Phe^6^, β-Ala^11^, Sta^13^, Nle^14^]BBN (6-14). The bombesin analogues have been published in the works of different research groups throughout many years, however, a direct comparison between them in terms of drug delivery has never been done.

Once the targeting peptides bind to receptors that are overexpressed on the surface of cancer cells, such as GRP-R, the PDC can be carried inside the cells via receptor-mediated endocytosis. Briefly, the receptor–ligand complex internalises via a special, coated vesicle that later fuses with an early endosome and matures into a late endosome. After the late endosome fuses with a lysosome, the PDC is also digested, the drug is released into the cytoplasm and binds to its target compartment inside the cell. To ensure this release, we have decided to insert cathepsin B cleavable tetrapeptide linkers, namely either GFLG or LRRY, between the homing peptide and Dau. Indeed, cathepsin B is highly expressed by lysosomes and can cleave between Gly-Phe and Leu-Arg [48,49], liberating the active metabolites Dau=Aoa-Gly-OH or Dau=Aoa-Leu-OH. Dau is part of the family of anthracyclines and elicits its cytostatic activity by intercalating between DNA base pairs. This prevents the topoisomerase II from resealing the DNA double helix, resulting in reduced cell proliferation. Its attachment to a peptide can be easily performed through an oxime linkage in a very simple and straightforward reaction that occurs between an aminooxy moiety and the C13 ketone on Dau (Figure 1). The high yields of the reaction are advantageous, especially when it comes to the production of an increased amount of conjugate for in vivo studies. Furthermore, thanks to its intrinsic fluorescence, it is possible to evaluate the internalisation of the Dau-conjugates by fluorescence-activated cell sorting (FACS) and confocal laser scanning microscopy (CLSM), without the need to produce alternative peptide conjugates [50]. To the best of our knowledge, so far, the delivery of a payload by bombesin-related peptides has only been studied using fluorescent molecules such as TAMRA, radioligands or nanocarriers. Drugs were attached to the peptides less often, and mainly using the original or truncated bombesin sequence instead of more promising analogues, or ones that are non-selective for GRP-R [22,43,44,47,51]. Oppositely, investigating features such as the cellular uptake directly, through a conjugate attached to a chemotherapeutic, gives the most realistic picture of the success of a homing peptide acting as a drug carrier. Thus, the generated structures provide valuable tools for the selection of targeting peptides, aiming to the further development of new tumour-selective conjugates containing different linker-payload systems.

We aimed to explore the targeting abilities of bombesin-related peptides to decide which sequence was the most suitable for the design of a successful DDS. Therefore, we studied features such as the cellular uptake, internalisation and cytostasis on selected cancer cell lines (human prostate cancer PC-3, and human breast cancer MDA-MB-231 and MDA-MB-453), the ability to release the cytotoxic payload in the lysosomal environment and the stability of our PDCs in mouse plasma. Moreover, the mRNA and protein expression levels of the receptor were quantified via RT-qPCR and Western blot, respectively. The best-performing conjugates were selected for further in vivo studies in tumour-bearing mice to evaluate their safety and tumour growth inhibition.

## 2. Results

### 2.1. Synthesis of the Peptide–Drug Conjugates

The peptide–drug conjugates contained either the native BBN (7-14), [D-Phe^6^]BBN (6-14) or four other BBN-analogues, based on the substitution of positions 11, 13 and 14. All targeting peptides were conjugated to daunorubicin through the GFLG (conjugates **G1**-**G5**) or the LRRY (conjugates **L1**-**L6**) cathepsin B cleavable linker (Figure 1).

Peptide sequences were synthesised via solid phase peptide synthesis, using the Fmoc/tBu strategy (Figure 1). The cleavage and contemporary side chain deprotection were performed using an appropriate TFA–scavenger mixture, depending on the presence of either the Gly-Phe-Leu-Gly or Leu-Arg-Arg-Tyr sequence, then purified by RP-HPLC (see Materials and Methods). The >=Aoa at the *N*-terminus, untouched after the cleavage, was then deprotected from the isopropylidene group using 1 M methoxyamine in 0.1 M NH_4_OAc buffer, pH 5. At this point, the peptides were conjugated to Dau via oxime linkage between the aminooxy moiety and the ketone at the C13 of the drug, and purified by RP-HPLC. The oxime bond was chosen because of its high stability in a large range of pHs. To ensure the purity of the final products, and in particular the absence of free Dau, the conjugates were characterised by analytical RP-HPLC and mass spectrometry (Table 1, Appendix A).

### 2.2. GRP-R Expression in Selected Cell Lines

The cancer cell lines used to test our newly developed PDCs were selected based on literature search. To verify our cell line selection, we investigated and quantified the relative mRNA expression of three bombesin receptors: GRP-R, NMB-R, and BRS-3. The expression of these genes was investigated in three human cancer cell lines: PC-3 (prostate), MDA-MB-453 and MDA-MB-231 (breast). The relative GRP-R levels were calculated using human β-actin as a reference gene. All cell lines expressed detectable levels of the GRP-R mRNA, but no signal was measured when using primers against NMB-R and BRS-3 mRNA molecules. When comparing the relative expression of GRP-R, MDA-MB-231 showed the lowest expression among the three cell lines. MDA-MB-453 and PC-3 exhibited 1.8- and 2.9-times higher expression, respectively (see Appendix A).

To further validate our cell line selection, we also measured the relative protein levels of GRP-R in the three investigated cell lines via Western blot analysis. We confirmed that all three cell lines express the receptor: a specific signal with a molecular weight of ~40 kDa is displayed (Appendix A). Considering the signal for the loading control β-actin, MDA-MB-453 cells show the highest expression of GRP-R, followed by PC-3 and MDA-MB-231 cells.

### 2.3. In Vitro Cytostatic Effect of Dau-BBN(7-14) Conjugates

The cytostatic effect of the conjugates and the free peptides was evaluated on the three mentioned human cancer cell lines expressing GRP-R. The free Dau was used as a positive control for comparison purposes; it displays an IC_50_ in the high nanomolar range.

Overall, the conjugates containing the LRRY spacer have a higher cytostatic effect than the ones with the GFLG spacer (Table 2). **L1**, which has the original BBN (7-14) sequence, and **L5**, bearing a D-Phe in position 6 and the Sta^13^–Leu^14^ bond at the C-terminus, shows the best activity in all three cell lines, with IC_50_ values in the low micromolar range. The Gly^11^/β-Ala^11^ and Leu^13^-Met^14^/Sta^13^-Nle^14^ substitutions, which led to a new BBN (6-14) peptide sequence, held by the conjugate **L6,** affect the activity only slightly. Contrarily, the two free peptides, bearing the sequences of **L5** and **L6**, do not show any activity on any of the cell lines.

The lack of toxicity of the conjugates on healthy cells was checked on MRC-5 human fibroblasts, proving that they are non-toxic on non-cancerous cells (see Appendix A).

### 2.4. Cellular Uptake and Localisation of Dau–BBN (7-14) Conjugates

We quantified the ability of the produced PDCs to promote the internalisation of daunorubicin after binding to GRP-R by flow cytometry. Each bioconjugate was incubated for 1.5 h in four concentrations (25 μM, 12.5 μM, 6.25 μM and 3.125 μM, see Appendix A), with the three cell lines that we have also used for the evaluation of the cytostatic effect. For better comparison, the uptake was described as the necessary concentration to internalise 50% of the compound (UC_50_) [52] (Table 3).

The uptake reflects the GRP-R protein expression: the lower UC_50_ values, hence the highest uptakes, are noticed in the human breast cancer cell line MDA-MB-453, whereas they are comparable in the two other cell lines.

As far as the internalisation of the individual conjugates is concerned, **L5** and **L6** are the most promising ones in these three cell lines. Notably, both hold the LRRY spacer and the Sta^13^. **L1** and **L2** have satisfactory UC_50_s, too. On the other hand, majority of the conjugates with the GFLG spacer are poorly internalised.

To visualise the intracellular localisation of the conjugates, confocal laser scanning microscopy (CLSM) was performed with a selected set of conjugates. PC-3 cells (as these cells were chosen for subsequent in vivo studies) were incubated with Dau conjugate peptides for 15, 45 and 90 min. Cells were fixed by 4% paraformaldehyde and nuclei were stained with Hoechst 33342. Imaging was performed on a Zeiss LSM 710 CLSM instrument. 

Although CLSM is not a quantitative method and laser intensities had to be adjusted to be able to visualise all the studied conjugates, it is visible from the signal-to-noise ratio that **L1** and **L2** internalise at a lower extent compared to **L5** and **L6**. While **L1**, **L5** and **L6** can be detected both in the cytoplasm, in vesicular structures (assumed to be lysosomes) and in the nuclei, **L2** is mostly visible in the cytoplasm. We could detect an enhanced fluorescent signal with the increment of incubation time in all cases (Figure 2).

### 2.5. Metabolism and Release of Dau=Aoa-Leu-OH in Lysosomal Environment

All the conjugates contain a cathepsin B cleavable spacer, either GFLG or LRRY, to allow the release of the toxic payload in lysosomes after receptor-mediated internalisation. We explored the metabolism of **L1**, **L3**, **L5** and **L6** in rat liver lysosomal homogenate to mimic this situation. The experiment was performed over 72 h, sampling at 0 min, 15 min, 30 min, 1 h, 2 h, 6 h, 24 h and 72 h in triplicates. All the (Dau=Aoa-LRRY)–BBN (7-14) conjugates have a similar degradation profile and they quickly release the active metabolite Dau=Aoa-Leu-OH, which can be detected after 15 min (Figure 3). As far as the degradation is concerned, **L1**, which contains only natural amino acids, is the one that degrades in the fastest way. Its signal cannot be detected anymore already after 1 h.

### 2.6. Stability of Dau–BBN (7-14) Conjugates

To confirm that the conjugates remain intact under in vitro testing conditions, and undesirable degradation products are not present to affect their internalisation or activity, we tested the stability of the three best-performing bioconjugates, **L1**, **L5** and **L6,** in the cell culture medium used for the cellular uptake and MTT assays. 

The conjugates were incubated in incomplete DMEM for 3 h (similarly to the internalisation studies) or in 2.5% FBS containing DMEM for 24 h (conditions for the MTT assay). All three PDCs were found to be highly stable in incomplete DMEM for 1.5 h, which is the incubation time used for the internalisation assay. No significant loss of **L1** and **L5** (<20%) or **L6** (<30%) was observed after 3 h (see Appendix A). 

The conjugates also proved to be stable in FBS containing DMEM, the medium used for the MTT assays. In this case, more than 85% of intact conjugates were detected after 6 h, and their concentration decreased only slightly between 6 h and 24 h; more than 65% of conjugate remained intact at 24 h. However, neither the free Dau nor the active Dau=Aoa-Leu-OH were detected (Appendix A).

Due to their low IC_50_ values and internalisation on the tested cell lines, conjugates **L1**, **L5** and **L6** were chosen for investigation of their stability in mouse plasma, as potential candidates for in vivo studies. They were incubated at a concentration of 10 µM, with mouse plasma for 24 h at 37 °C. Samples were collected at 0 h, 30 min, 1 h, 2 h, 4 h, 8 h and 24 h and analysed by HPLC-MS. **L5** and **L6** are noticeably more stable than **L1** (Figure 3): whereas more than 80% of the former are still intact after one day, the detected fraction for the latter diminished more rapidly, to 20% after 8 h (t_1/2_ = 3.8 h). Furthermore, after 2 h, **L1** starts to release the toxic metabolite Dau=Aoa-Leu-OH, whereas this happens only after 24 h for **L5** and **L6**, in a considerably low amount (see Appendix A).

### 2.7. In Vivo Chronic Toxicity Studies of L5 and L6

Based on the stability assay performed in murine plasma, we decided to investigate **L5** and **L6** in vivo. To determine the dosage for the tumour growth inhibition experiment, we performed a chronic toxicity study using healthy mice. PDCs were administered every 5th day in three different doses calculated on the Dau content: 5 mg/kg, 10 mg/kg, and 20 mg/kg (Figure 4A). None of the PDCs resulted in critical weight loss at any dose until the end of the experiment; however, 20 mg/kg of **L5** and **L6** compounds induced a relatively high, 10–15%, decrease in mouse weight. When 10 mg/kg drug was applied, the fitness of mice was not affected more than that of mice that received 5 mg/kg. According to this data, we decided to apply a dose of 10 mg/kg when investigating the effect of **L5** and **L6** on in vivo tumour growth (Figure 4B).

### 2.8. In Vivo Tumour Growth Inhibition by L5 and L6

In vivo antitumour efficacy of PDCs was investigated using our murine xenograft model bearing PC-3 human prostatic adenocarcinoma (Figure 4B). Tumour inhibition of the conjugates was compared to mice receiving mock and free daunorubicin treatments. Although the free drug was administered in 1 mg/kg, it did not significantly reduce tumour growth. Moreover, even in such a low concentration, free daunorubicin decreased the fitness of mice to a critical state already at day 26 (see Appendix A). On the other hand, compounds **L5** and **L6** showed lower toxicity compared to free daunorubicin, even though the conjugates were applied at a much higher dose of 10 mg/kg. In the case of PDC-treated mice, the size of the tumours was not significantly smaller at the termination of the experiment compared to controls; however, **L5** and **L6** treatments resulted in 21.4% and 31.4% growth inhibition, respectively. When comparing tumour weights, **L5**-treated mice showed a decrease of 16.6%, whereas **L6**-treated mice exhibited 33.1% reduction compared to control animals at day 33.

Murine experiments are indisputably important for drug development, but the duration of the experiments is usually shorter than the treatment of human patients. To better understand the effect of the newly developed compounds, we calculated tumour doubling time (DT) of tumours. Based on these values, we can estimate which compounds would perform better in the case of a longer treatment. The comparison of DTs of tumours in the different groups shows that both **L5** and **L6** result in a significant increase compared to the treatment with free daunorubicin (Figure 4C). No significant difference was observed between controls and PDC-treated animals; however, DTs of tumours treated with **L5** and **L6** were 8.5% and 11.5% higher compared to the tumours harvested from control animals, respectively.

## 3. Discussion

The importance of drug delivery systems, such as drug conjugates for targeted tumour therapy, has become more evident in recent years. Exploiting a suitable target for drug delivery helps to develop an efficacious treatment while significantly reducing the side effects associated with chemotherapy. Among the possible targets, bombesin receptors, and in particular the mammalian GRP-R, have been widely studied, demonstrating its upregulation in several types of cancer cells and its function in promoting cell proliferation when activated [8,19,20,21,22]. A great effort has been made since the late 1990s to produce bombesin-related peptides that could directly antagonise this event, and later conjugates with radioligands or drugs to detect and fight malignancies. However, a direct comparison in terms of tumour targeting and drug delivery of the bombesin analogues developed throughout the years was never carried out. We collected many GRP-R-binding putative homing peptides, and conjugated them to the anthracycline daunorubicin through cathepsin B cleavable spacers and an oxime bond, aiming to select the most appropriate bombesin analogues for targeted drug delivery. Based on the information that we gathered from the in vitro investigation, we designed a new peptide sequence and used it to synthesise a new conjugate. We found that this compound, **L6**, together with **L5**, had appropriate features to be evaluated in vivo. 

The selection of suitable cell lines to evaluate our conjugates started by a literature search. The human prostate cancer cell line PC-3 and the human breast cancer cell lines MDA-MB-231 and MDA-MB-453 are widely used by research groups working with bombesin, as they are reported to express GRP-R [24,25,26,27,30,32,36,41,53,54,55]. Nonetheless, its cellular levels were rarely assessed. Therefore, we validated our cell lines by demonstrating the presence of this receptor and quantifying the expression levels by Western blot and qPCR. As a result, we could investigate the cellular uptake and the cytostatic activity of the eleven conjugates on these cell lines, expecting a receptor-mediated process. Because of the fluorescence of Dau, we could study the uptake by FACS and CLSM without changing the inherent properties of the compounds [50]. Overall, the uptake reflects the expression levels of GRP-R: a better internalisation of the conjugates is observed in the MDA-MB-453 cell line compared to MDA-MB-231 and PC-3. However, the cytostatic activity of the conjugates is not always higher in MDA-MB-453. We suppose that a difference in intracellular signalling pathways (possibly due to diverse levels of other proteins involved in cell proliferation-related processes) among the three cell lines can affect the way our compounds exert their activity. Consequently, a promising cellular uptake does not always precede a good IC_50_.

The conjugates **G3** and **L3**, and **G4** and **L4** contain as a homing [D-Phe^6^, β-Ala^11^, Aib^13^, Nle^14^]BBN (6-14) and [D-Phe^6^, β-Ala^11^, *N*-Me-Ala^13^, Nle^14^]BBN (6-14), respectively. These two peptides were reported as putative drug shuttles for targeted tumour therapy by Hoppenz et al. [27], with great features in terms of selectivity towards GRP-R and plasma stability. Nevertheless, our experiments show that, despite a similar cytostatic activity, the uptake of the conjugates containing such peptides is reduced compared to others in all the cell lines, particularly **L5** and **L6**. On the other hand, the two latter conjugates have as homing peptides [D-Phe^6^, Sta^13^, Leu^14^]BBN(6-14) and [D-Phe^6^, β-Ala^11^, Sta^13^, Nle^14^]BBN (6-14), respectively. The peptide in **L6** is a novel bombesin analogue that we have designed in our laboratory based on the successful targeting ability of the other statine-containing peptide, used for compounds **G5** and **L5**, which is widely described and that we confirm in this paper. Starting from this, we have decided to maintain Nle^14^ as in most of the bombesin analogues in literature, and to substitute ^11^Gly with β-Ala, which is reported in both non-selective and GRP-R-selective ligands that have been the starting points for peptides with an increased affinity and stability [27,56]. Moreover, given the significantly improved in vitro activity of **L5** compared to **G5**, we have preferred to use the LRRY spacer for **L6**. The presence of Sta^13^-Leu^14^ at the C-terminus is reported to confer antagonistic properties to bombesin-like peptides [28,31,57], and we may imply that the replacement of Leu by Nle does not modify the activity. It was also previously reported that the interaction between GRP-R and peptides having an antagonistic character would not promote the internalisation of the peptide–receptor complex [39,58,59]. Surprisingly, however, we observed that **L5** and **L6** are the best internalised conjugates in both the prostate and breast cancer cell lines. In the case of **L5**, this is flanked by the lowest IC_50_ values in the three tested cell lines among all the conjugates, with a value as low as 2.22 ± 0.19 μM in the human prostate cancer cell line PC-3. The homing peptides used in compounds **G1**/**L1** and **G2**/**L2** are, respectively, the standard truncated BBN (7-14), QWAVGHL-Nle, and the alternative with D-Phe^6^. Among these four, **L1** is the only one with promising features in terms of uptake and cytostatic activity in all three cell lines. Furthermore, the images obtained by the confocal microscope reveal that **L2** does not reach the nucleus after being internalised by PC-3, oppositely to **L1**, **L5** and **L6**. As a result, **L1**, **L5** and **L6** were considered for further biochemical assays.

The reliability of the results of the in vitro biological assays was demonstrated by the stability of the conjugates in cell culture medium, free from or containing FBS: we show that, despite the detected amount of the compounds slightly decreasing after 3 h and 24 h, respectively, no free Dau or Dau-containing metabolites are released. Therefore, the observed IC_50_ values and the cellular uptake are not affected.

The specific release of the drug from the conjugates in the tumour environment was ensured by inserting cathepsin B cleavable spacers, namely either the tetrapeptide GFLG or LRRY, between the homing peptide and the drug [48,49]. Given that this enzyme is overexpressed in tumour cells by the lysosomes, the premature release of the toxin in the bloodstream should be prevented, whereas it should occur in the cancerous cells, after the receptor-mediated endocytosis, triggered by the binding between targeting peptide and GRP-R. Additionally, to reduce the possibility of a loss of the toxic payload in blood circulation even further, an aminooxyacetic acid was added at the *N*-terminus of the spacers, to which Dau was attached via an oxime bond. This chemistry provides a better chemical and enzymatic stability compared to other linkage strategies, such as esters or hydrazones [8,60,61]. On the other hand, it does not allow for the release of the free drug even after penetration into the lysosomal environment. However, another active metabolite is released due to the cathepsin-mediated cleavage between either Gly and Phe or Leu and Arg, respectively: Dau=Aoa-Gly-OH or Dau=Aoa-Leu-OH. Although the activity is generally lower than that of the free Dau, it has been proved that even small Dau-containing metabolites can intercalate DNA and reduce cell proliferation [61,62]. To confirm our hypotheses, we have studied the degradation of **L1**, **L3**, **L5** and **L6** in rat liver lysosomal homogenate. Oppositely to **L1**, **L5** and **L6**, conjugate **L3** showed a surprisingly low cytostatic activity in vitro. Hence, it was selected to check whether this was due to a poor degradation of the conjugate in a lysosomal environment or the low uptake. All four conjugates showed similar properties: they degraded rapidly and released the toxic metabolite in less than 30 min. Therefore, we deduce that the poor cytostatic activity of **L3** is caused by its weaker internalisation, as opposed to the improved activity and uptake of **L1**, **L5** and **L6**. 

Unfortunately, due to poor solubility in the conditions of the assay, we could not assess the metabolism of any of the compounds containing the GFLG spacer.

Our last concern before the in vivo evaluation of the conjugates was their stability in circulation. Therefore, we incubated **L1**, **L5** and **L6** in mouse plasma to explore whether their half-life was suitable to reach the target. We assumed that sampling for a time span longer than 24 h would not be necessary, since PDCs are generally excreted through the bladder after 6–8 h. While more than 80% of both **L5** and **L6** could be detected after 24 h, a t_1/2_ = 3.8 h was calculated for **L1**. Despite such a half-life being considered appropriate for tumour targeting, we decided to exclude this bioconjugate from in vivo testing because we could detect a minor release of the toxic metabolite Dau=Aoa-Leu-OH after 2 h (see Appendix A), preferring the more stable **L5** and **L6**.

Before testing the effect of our newly developed PDCs on tumour growth in vivo, we performed a chronic toxicity study to determine the ideal dosage of the compounds. Administration of daunorubicin is shown to cause severe cardiotoxic side effects, therefore, its maximum tolerated dose is 1 mg/kg in the case of our experimental model [17]. However, when conjugated to targeting moieties, its side effects can be reduced; moreover, its tumour inhibition capabilities can be increased. We showed that healthy mice do not suffer from critical weight loss even at a dose of 20 mg/kg daunorubicin, when the drug is conjugated to the newly developed targeting peptides. On the other hand, we decided to reduce the dosage of 10 mg/kg, due to the comparable weight loss and general conditions of mice, to 5 mg/kg, which was the lowest tested daunorubicin content.

After determining the dosage, we established our murine xenograft model by inoculating PC-3 cells subcutaneously into NOD-SCID mice. **L5** and **L6** did not result in a significant inhibition in tumour size compared to the control; however, we reported a considerable 21.4% and 31.4% decrease, respectively. Data obtained from in vivo experiments often have high standard deviation; therefore, even a relatively high inhibition in tumour growth can turn out to be non-significant. To overcome this issue, it is always possible to include more participants in the experiment, but this would not be in agreement with the 3R guidelines [63]. Nevertheless, normalising our data sets by calculating the DTs of the individual tumours, is a way to resolve the problems of high standard deviation while, at the same time, designing experiments according to the 3Rs. When we compared the DTs of the tumours obtained from the different groups, we were able to show a significant increase between PDC-treated and free daunorubicin-treated groups. This data indicate that the growth rate of the tumours is significantly reduced in the case of **L5**, and **L6** treatment compared to the free drug. Therefore, we showed that the attachment of our newly developed targeting moieties not only reduces the toxicity of the free drug, but it also increases its antitumoural activity. 

In summary, we produced eleven conjugates composed of a bombesin analogue, attached to Dau via cathepsin B cleavable linkers, and compared them in terms of cytostatic activity and cellular uptake in cancer cells, ability to release the payload in a lysosomal environment, stability in cell culture medium and mouse plasma. To the best of our knowledge, this was the first direct comparison of such an array of bombesin-related peptides for tumour drug delivery. This led to the selection of two compounds, namely **L5** and **L6**, for further in vivo studies. Finally, using our in vivo experimental model, we showed that **L5** and **L6** increase the selectivity and decrease the toxicity of the free drug. Having proved that bombesin-based Dau conjugates are helpful tools to target cancer cells, our next aim is to generate novel cathepsin-labile conjugates with the homing peptides of **L5** and **L6,** and a more potent toxic payload, such as auristatins, to achieve a stronger efficacy while maintaining their selectivity.

## 4. Materials and Methods

### 4.1. Chemical Reagents

Rink-amide MBHA resin and all amino acid derivatives were purchased from Iris Biotech GmBH (Marktredwitz, Germany), except for *N*-Fmoc-L-Statine, which was obtained from Fluorochem Ltd. (Hadifield, UK). Aminooxyacetic acid, scavengers, coupling agents (1-hydroxybenzotriazole hydrate (HOBt), *N*,*N*′-diisopropylcarbodiimide (DIC)), and cleavage reagents (triisopropylsilane (TIS), piperidine, 1,8-diazabicyclo(5.4.0)undec-7-ene (DBU), 1,2-ethandithiol (EDT), thioanisole), diisopropylethylamine (DIPEA) and acetic anhydride (Ac_2_O) were purchased from Sigma Aldrich Kft. (Budapest, Hungary). Daunorubicin hydrochloride was provided by IVAX (Budapest, Hungary). *N*,*N*-Dimethylformamide (DMF), dichloromethane (DCM), diethyl ether (Et_2_O), trifluoroacetic acid (TFA) and HPLC grade acetonitrile (MeCN) were purchased from VWR Chemicals (Debrecen, Hungary). All reagents and solvents were of analytical grade or highest available purity.

### 4.2. Synthesis of Peptide Sequences

All the peptides were synthesised manually on a Rink-amide MBHA resin (loading capacity: 0.69 mmol/g—Iris Biotech) with a Fmoc/tBu strategy. Each coupling was performed with 3 eq of amino acid, 3 eq of HOBt and 3 eq of DIC, for 1 h. The final coupling of the isopropylidene-protected aminooxyacetic acid (≥Aoa-OH) was performed with 3 eq of Aoa derivative, 3 eq of HOBt and 9 eq of DIC, for 1 h. The peptides containing the GFLG spacer were cleaved from the resin with a cocktail composed of TFA, TIS and dH_2_O (95:2.5:2.5 *v*/*v*%), for 1.5 h, RT. The ones containing the LRRY spacer were cleaved by adding the cleavage reagents TFA, crystalline phenol, thioanisole, dH_2_O, EDT (85:6:4:4:1 *v*/*m*/*v*/*v*/*v*%), for 2 h, RT. Peptides were precipitated in a 10× excess of ice-cold Et_2_O. The precipitates were then centrifuged and washed three times (3 min, 4400rpm Eppendorf Centrifuge 5702) with fresh Et_2_O, dissolved in dH_2_O:MeCN (0.1% TFA) 1:1 (*v*/*v*%) and freeze-dried.

### 4.3. Isopropylidene Deprotection and Conjugation to Daunorubicin

The isopropylidene deprotection was performed by 1 M CH_3_ONH_2_ in 0.1 M NH_4_OAc buffer (pH 5) and the smallest amount of DMF necessary to dissolve the peptide, for 1–2 h, RT. A purification step by RP-HPLC is always needed before the conjugation to the daunorubicin to get rid of the excess of CH_3_ONH_2_. To perform the conjugation, 1.3 eq of Dau and 10 mg/mL freeze-dried peptide were dissolved in 0.1 M NH_4_OAc buffer (DMF), pH 5, and stirred o/n, RT. The peptide–Dau bioconjugates were purified via RP-HPLC either on a Luna C18 or Jupiter C4 column and the product-containing fractions were combined and freeze-dried. The final compounds were characterised by analytical RP-HPLC (Aeris C18 column) and ESI-MS.

### 4.4. RP-HPLC

The crude peptides and the bioconjugates were purified on a KNAUER 2501 HPLC system (H. Knauer, Bad Homburg, Germany) using either a preparative Phenomenex Luna C18(2) column (100 Å, 10 μm, 250 × 21.2 mm) (Torrance, CA, USA) or a preparative Phenomenex Jupiter C4 column (300 Å, 10 μm, 250 × 21.2 mm). A linear gradient elution (0 min 20% B; 5 min 20% B; 10 min 30% B; 50 min 75% B) with eluent A (0.1% TFA in water) and eluent B (0.1% TFA in MeCN/H_2_O (80:20, *v*/*v*)) was used at a flow rate of 14 mL/min. Peaks were detected at 220 nm. Analytical RP-HPLC was performed on a KNAUER Azura 2.1S HPLC system using a Phenomenex Aeris C18 column (100 Å, 3.6 μm, 250 × 4.6 mm) as a stationary phase. A linear gradient elution (0 min 2% B; 5 min 2% B; 30 min 90% B) at a flow rate of 1 mL/min was used with the eluents described above. Peaks were detected at 220 nm.

### 4.5. Liquid Chromatography–Mass Spectrometry (LC–MS)

Electrospray ionization (ESI) mass spectrometric analyses were carried out on a Q Exactive^TM^ Focus, high-resolution and high-mass accuracy, hybrid quadrupole-orbitrap mass spectrometer (Thermo Fisher Scientific, Bremen, Germany). Spectra were acquired in the 200–2000 *m/z* range. Samples were dissolved in a mixture of MeCN/water (1:1, *v*/*v*) and 0.1% formic acid. Liquid chromatography–mass spectrometry (LC–MS) was carried out on an UltiMate 3000 UHPLC system (Thermo Fisher Scientific) coupled to the same spectrometer. Compounds were separated on a Supelco Ascentis C18 column (90 Å, 150 × 2.1 mm, 3 µm) (Hesperia, CA, USA), using a linear gradient from 20–90% B in 20 min (eluent A: ddH_2_O, 0.1% HCOOH; eluent B: 80% MeCN, 0.1% HCOOH at a flow rate of 0.2 mL/min) and the column temperature was set to 40 °C. High-resolution mass spectra were acquired in the 200–2000 *m/z* range. LC–MS data were analysed by the Xcalibur^TM^ software (Thermo Fisher Scientific).

### 4.6. Cell Lines and Culturing

In vitro biological effects of the compounds were studied on MDA-MB-231 human breast adenocarcinoma [64], PC-3 human prostate adenocarcinoma [65], and MDA-MB-453 human metastatic epithelial breast carcinoma [66] cells. Cell lines were generous gifts of Dr. József Tóvári (Department of Experimental Pharmacology, National Institute of Oncology, Budapest, Hungary). MDA-MB-231 and MDA-MB-453 cells were cultured in DMEM medium (Lonza, Basel, Switzerland), supplemented with 10% FBS (EuroClone, Pero, Italy), 2 mM L-glutamine (BioSera, Nuaille, France), penicillin–streptomycin antibiotics mixture (50 IU/mL and 50 μg/mL, respectively), 1 mM sodium pyruvate (both obtained from Lonza, Basel, Switzerland), and 1% non-essential amino acid mixture (BioSera, Nuaille, France). PC-3 cells were cultured in RPMI-1640 medium (Lonza, Basel, Switzerland) supplemented with 10% FBS (EuroClone, Pero, Italy), 2 mM L-glutamine (EuroClone, Pero, Italy), and penicillin–streptomycin antibiotics mixture (50 IU/mL and 50 μg/mL, respectively) (Lonza, Basel, Switzerland). The cultures were maintained at 37 °C in a humidified atmosphere with 5% CO_2_. The cells were grown to confluency before use.

### 4.7. MTT Assays

Cells were distributed into 96-well tissue culture plates with an initial cell number of 5.0 × 10^3^ per well. After 24 h incubation at 37 °C, the cells were treated with the compounds in 200 µL final volume containing 1.0 *v*/*v*% DMSO and 10% H_2_O. The cells were incubated with the compounds at 3.125–25 µM concentration range for 24 h, whereas control cells were treated with serum-free medium only or with DMSO (c = 1.0 *v*/*v*%) and H_2_O (c = 10.0 *v*/*v*%) at 37 °C for 24 h. After incubation, the cells were washed twice with a serum-free medium. To determine the in vitro cytostatic effect, the cells were further cultured up to 72 h in a 10% serum-containing medium. A solution of 3-(4,5-dimethylthiazol-2-yl)-2,5-diphenyltetrazolium bromide and MTT-solution, (45 µL, 2 mg/mL, final concentration: 0.37 µg/mL) were added to each well. The respiratory chain [67] and other electron transport systems [68] reduce MTT, and thereby form non-water-soluble violet formazan crystals within the cell [69]. The amount of these crystals can be determined spectrophotometrically and serves as an estimate for the number of mitochondria, and hence the number of living cells in the well [70]. After 3 h of incubation, the cells were centrifuged for 5 min at 1000 g and the supernatant was removed. The obtained formazan crystals were dissolved in DMSO (100 μL) and the optical density (OD) of the samples was measured at λ = 540 and 620 nm, respectively, using ELISA Reader (iEMS Reader, Labsystems, Finland). OD_620_ values were subtracted from OD_540_ values. The percentage of cytostasis was calculated by using the following equation:Cytostatic effect (%) = [1 − (OD_treated_/OD_control_)] × 100(1)

Values of OD_treated_ and OD_control_ correspond to the optical densities of the treated and the control cells, respectively. In each case, two independent experiments were carried out with four parallel measurements. The 50% inhibitory concentration (IC_50_) values were determined from the dose–response curves. The curves were defined using Microcal™ Origin 2018 software: cytostasis was plotted as a function of concentration, fitted to a sigmoidal curve, and based on this curve, the half-maximal inhibitory concentration (IC_50_) value was determined. IC_50_ represents the concentration of a compound that is required for 50% inhibition in vitro and is expressed in micromolar units.

### 4.8. Stability of Bombesin-Based Bioconjugates in Mouse Plasma

The conjugates were dissolved in ddH_2_O containing 10% DMSO (stock solution: 100 μM) and diluted with mouse plasma to a final concentration of 10 μM. The mixture was incubated and stirred at 37 °C, 600 rpm, and 30 µL aliquots were collected after 0 min, 0.5, 1, 2, 4, 8 and 24 h, in triplicates. Then, to each plasma sample, 90 µL of 0.1% formic acid containing MeCN was added, and the samples were frozen at −25 °C for ≥30 min to aid protein precipitation. Large plasma proteins were removed by centrifugation (15 min, 5000× *g*, 4 °C). Aliquots of 50 µL supernatant were diluted with an equal volume of ddH_2_O + 0.1% formic acid and analysed by LC–MS. Control measurements were performed in the same manner (10 μM bioconjugate in 100% ddH_2_O).

### 4.9. Stability of Bombesin-Based Bioconjugates in Cell Culture Media

DMSO stock solutions of the conjugates (2.5 mM) were diluted in DMEM free from or containing 2.5% FBS, to a final concentration of 25 μM. The mixture was incubated and stirred at 37 °C, 600 rpm, and 30 µL aliquots were collected after 0 min, 0.5, 1, 2, 4, 8 and 24 h, in duplicates. Then, to each sample, 90 µL of 0.1% formic acid containing MeCN was added and the samples were frozen at −25 °C for ≥30 min to aid protein precipitation. Large FBS components were removed by centrifugation (15 min, 5000 g, 4 °C). Aliquots of 50 µL supernatant were diluted with an equal volume of ddH_2_O + 0.1% formic acid and analysed by LC–MS. Control measurements were performed in the same manner (25 μM bioconjugate in 100% ddH_2_O).

### 4.10. Isolation of Lysosomes from Rat Liver

The rat liver lysosomal homogenate was prepared based on our previously published procedure [62], with small modifications: livers from two male Wistar rats (32.6 g and 41.2 g, respectively) were collected, minced with a knife and homogenised in two volumes of ice-cold 0.3 M sucrose, using a T25 Ultra Turrax homogenizer (∼100 mL). The homogenate was diluted with one volume of 0.3 M sucrose (200 mL). The nuclei and cell debris were centrifuged at 700 g for 10 min at 4 °C (Sorvall LYNX 4000 centrifuge with Fiberlite™ F21-8 × 50y rotor, Thermo Scientific). The pellet was washed with three volumes of 0.3 M sucrose solution and centrifuged again at 700 g for 10 min at 4 °C. After that, the post-nuclear supernatant (PNS, Σ 250 mL) was centrifuged at 10,000× *g* for 10 min at 4 °C to sediment the crude lysosomal–mitochondrial fraction. The sediment was re-homogenized in 40 mL of 0.3 M sucrose containing CaCl_2_ (final CaCl_2_ concentration: 1 mM). The homogenate was incubated at 37 °C for 5 min for the mitochondria swallowing, and then mixed with 40 mL of 50% Percoll (20 mL Percoll + 20 mL 0.6 M sucrose solution) added to the solution, followed by the centrifugation of the homogenate at 20,000× *g* for 20 min at 4 °C (centrifugation deacceleration: 3 was applied to avoid the reorientation of the Percoll gradient). The supernatant was removed, while the pellet was resuspended in 5–10 mL 0.3 M sucrose and centrifuged again at 20,000× *g* for 20 min at 4 °C (centrifugation deacceleration: 3 was applied). The hard brown pellet was the lysosomal fraction, which was further diluted 1:1, with 0.3 M sucrose for easier pipetting. The protein concentration was determined by measuring the absorbance at 260 nm and 280 nm, and was calculated using the formula: Concentration (μg/μL) = 1.55 ∗ A_280_ − 0.76 ∗ A_260_; it was determined as 71.76 μg/μL. The activity of the lysosomal homogenate was validated in comparison with the previously prepared homogenate by performing LC–MS-based degradation assay with selected lysosomally cleavable linker-containing bioconjugates of the research group. Based on the results, the conjugate–lysosomal protein ratio was accordingly adjusted. 

### 4.11. Metabolism in Rat Liver Lysosomal Homogenate

The conjugates (2.5 μg/μL in DMSO, 5 μL) were diluted with 0.2 M NaOAc solution (pH 5.05, 495 μL) to 0.025 μg/μL. Lysosomal homogenate (8.6 μL) was diluted with 66.4 μL 0.2 M NaOAc solution (pH 5.05), to have a protein concentration of 8.27 μg/μL. To start the assay, 15 μL lysosomal homogenate (8.27 μg/μL) was added to 500 μL conjugate solution (0.025 μg/μL), to have a 1:10 (*w*/*w*) conjugate–lysosomal protein ratio. Furthermore, a control reaction mixture was prepared, containing the conjugate (2.5 μg/μL in DMSO, 5 μL) and 510 μL of 0.2 M NaOAc solution (pH 5.05). The samples were stirred at 600 rpm at 37 °C and 50 µL aliquots were taken out at 0 min, 15 min, 30 min, 1 h, 2 h, 6 h, 24 h, and 72 h, in triplicates. The enzymatic activity was quenched by adding 5 μL formic acid to the samples. After this procedure, samples were frozen immediately at −25 °C. Control samples were collected at 0 min, 1 h, 6 h, 24 h and 72 h. The composition of the samples was determined by HPLC-MS.

### 4.12. RT-qPCR Measurements

The relative RNA expression of GRP-R, NMB-R and BRS3 genes was determined using the RT-qPCR method. The total RNA from each cell line was isolated using the Trizol^®^ reagent (Ambion, by Life Technologies, Carlsbad, CA, USA). The purity and concentration of the extracted RNA samples were measured using a spectrophotometer (NanoDrop ND-1000, Wilmington, DE, USA) at an absorbance of 260 nm and 280 nm. Synthesis of cDNA was carried out in Eppendorf 5331 Mastercycler Gradient thermocycler (Eppendorf, Enfield, CT, USA). Five hundred nanograms of total RNA was used according to the protocol of the Reverse Transcription System provided by Promega (Promega, Madison, Wisconsin, USA). After reverse transcription, the cDNA samples were stored at −20 °C until further processing. Primers were obtained from Sigma-Aldrich, St. Louis, MO, USA, and were designed based on the reference sequences obtained from NCBI RefSeq (GRP-R—NM_005314; NMB-R—NM_002511.4, NM_001324307.2, NM_001324308.2; BRS3—NM_001727). Primer sequences are as follows: GRP-R_forward—gggagacctgctcctcctaa, GRP-R_reverse—gccgtgagtgtgaagacaga, NMB-R_forward—gaacccacagaagacccgag, NMB-R_reverse—catccttacccgcctccaag, BRS-3_forward—caaggcagttgtgaagccac, BRS3_reverse—aaacagagccaccaacacca. Measurements were done using SsoAdvanced Universal SYBR^®^ Green Supermix assay (Bio-Rad, Hercules, CA, USA) with a CFX96 Touch Real-Time PCR Detection System (Bio-Rad, Hercules, CA, USA). Relative expression was determined by normalising the expression levels to the lowest expressing cell line, human breast cancer cell line MDA-MB-231. Data represent three independent experiments, each performed in triplicates.

### 4.13. Western Blot

For quantification of GRP-R protein expression, whole-cell lysates were prepared. Briefly, cells were lysed in lysis buffer (25 mM Tris pH 7.4, 150 mM NaCl, 1 mM TCEP, 2 mM EDTA, 1% Triton-X 100, 1:100 Protease Inhibitor Cocktail, Halt™ Protease Inhibitor Cocktail, Thermo Fisher) for 35 min, at 4 °C. Lysates were centrifuged at 13,500 rpm for 30 min and the supernatant was stored at −80 °C. The concentration of the lysates was measured by Qubit Protein Assay Kit (Thermo Fisher). An equal amount of proteins was loaded to 10% Tris-tricine gel for SDS-PAGE and blotted to polyvinylidene fluoride (PVDF) membrane with Bio-Rad Wet Blotting System (Bio-Rad Hungary, Budapest, Hungary). GRP-R was detected by anti-GRP-R antibody (rabbit polyclonal antibody, PA5-26791, 1:1000, Invitrogen, Waltham, MA, USA) and anti-rabbit HRP-conjugated secondary antibody (1:2000, sc-2004, Santa Cruz Biotechnology, Dallas, TX, USA). As GRP-R and the loading control β-actin have similar molecular weights, the membrane was stripped by a stripping buffer (20 mM glycine pH 2.2, 0.1% SDS, 0.1% Tween-20). β-actin was detected by anti-actin antibody (goat polyclonal antibody, sc-1616, 1:2000, Santa Cruz Biotechnology) and anti-goat HRP-conjugated secondary antibody (sc-2354, 1:2500, Santa Cruz Biotechnology). A chemiluminescent signal was detected after the addition of enhanced chemiluminescence (ECL) substrate (SuperSignal West Pico PLUS Chemiluminescent Substrate, Thermo Fisher Scientific), by ChemiDoc XRS+ Detection System (Bio-Rad Hungary).

### 4.14. Flow Cytometry

Cells were plated into a 24-well tissue culturing plate, with the initial cell number of 10^5^ cells/well 24 h before the experiment. Cells were washed first with serum-free medium, and then incubated with the compounds dissolved in serum-free medium at 3.125 g/mL, 6.25 g/mL, 12.5 g/mL and 25 g/mL concentrations for 90 min at 37 °C. Control cells were treated with serum-free medium at the same conditions. Following that, cells were washed once with serum-free medium and then once with HPMI buffer (9 mM glucose, 10 mM NaHCO_3_, 119 mM NaCl, 9 mM HEPES, 5 mM KCl, 0.85 mM MgCl_2_, 0.053 mM CaCl_2_, 5 mM Na_2_HPO_4_ × 2H_2_O, pH 7.4 [71]), treated for 10 min with 0.25% trypsin (Merck, Darmstadt, Germany) and harvested. After that, HPMI supplemented with 10% FCS was added to the cells that were transferred to cytometry tubes (Sarstedt, Nümbrecht, Germany). Cells were then centrifuged at 1000 rpm for 5 min and the pellet was resuspended in HPMI. The cells were examined in a BD LSR II flow cytometer (Beckton Dickinson, Franklin Lakes, NJ, USA). Data were recorded from 5000–10,000 cells at ex = 488 nm and processed using the FACSDiva 5.0 software. Fluorescence mean was calculated, and besides that, the percent of daunomycin-positive cells was determined from the population of living cells. Statistical analysis of data was performed using a Student’s *t*-test at the 5% significance level (*p* < 0.05).

### 4.15. Confocal Microscopy

Cells were seeded to 24-well plates (5 × 10^4^ cells/well) that contained glass coverslips (diameter: 13 mm, VWR Hungary Kft.) one day before the experiment. Cells were treated with the conjugates (12.5 μM dissolved in serum-free medium) for 15 min, 45 min and 90 min. After washing with serum-free medium and PBS, cells were fixed with 4% paraformaldehyde (37 °C, 15 min, Sigma-Aldrich). Nuclei were stained with Hoechst 33342 (0.3 μg/mL, dissolved in PBS, 37 °C, 10 min, Thermo Fisher Scientific). After washing three times with PBS, coverslips were mounted to microscopy slides (VWR Hungary Kft.) using Mowiol 4-88 (Sigma-Aldrich). Images were captured and photographed using a ZeissLSM 710 confocal laser scanning microscope (Carl Zeiss AG, Jena, Germany), and processed by ZEN black software (Zeiss).

### 4.16. Materials and Animals

Adult male inbred NOD-SCID mice were bred in a specified pathogen-free (SPF) environment at the National Institute of Oncology (Budapest, Hungary). Mice were used in chronic toxicity studies and subcutaneous PC-3 human prostate tumour model experiments. Mice were kept in a sterile environment in Makrolon^®^ cages at 22–24 °C (40–50% humidity), with light regulation of 12/12 h light/dark. The animals had free access to tap water and were fed a sterilised standard diet (VRF1, autoclavable, Akronom Kft., Budapest, Hungary) ad libitum. Animals used in our study were taken care of according to the “Guiding Principles for the Care and Use of Animals” based on the Helsinki declaration, and they were approved by the ethical committee of the National Institute of Oncology. Animal housing density was according to the regulations and recommendations from directive 2010/63/EU of the European Parliament and of the Council of the European Union on the protection of animals used for scientific purposes. Permission license for breeding and performing experiments with laboratory animals: PEI/001/1738-3/2015 and PE/EA/1461-7/2020.

### 4.17. In Vivo Toxicity and Antitumour Efficacy of the Bombesin-Based Bioconjugates

The toxicity experiments were done in healthy NOD-SCID mice. Treatments were administered every 5th day. Three different concentrations were tested, 5 mg/kg, 10 mg/kg and 20 mg/kg, respectively. Each group consisted of three mice. Mouse weight was monitored five times per week.

The testing of the antitumour conjugates was carried out in a murine xenograft model. It was established by inoculating subcutaneously 100 µL of cell suspension of PC-3 prostate cancer cells with a concentration of 2 × 10^7^ cells/mL.

Once the tumour volume reached 60 mm^3^, mice were randomised and assigned to different groups for each treatment, respectively 0.9% saline as control, free daunorubicin, **L5**, and **L6**. Each group consisted of eight mice. Peptide–drug conjugates were injected intraperitoneally every 5^th^ day, five times in total (daunorubicin dosage of 10 mg/kg for each treatment). In the case of free daunorubicin, two treatments were administered at the same time points as the first two treatments with conjugates, with a dosage of 1 mg/kg, the maximum tolerated dose (MTD). Weight and tumour size of mice were monitored throughout the whole experiment, two times per week. On day 33, mice were euthanized, and primary tumour, heart, lung, liver and spleen were harvested and stored in 4% formalin (Molar Chemicals, Halásztelek, Hungary). After 2–3 days of incubation in formalin, tissue samples were examined macroscopically. Organs and primary tumours from the different treatment groups were weighed and inspected under a stereomicroscope to see any occurrent changes in morphology compared to the controls. 

### 4.18. Statistical Analysis

In vitro data are shown as mean ± standard deviation (SD), and in vivo data are presented as mean ± standard error of the mean (SEM). Comparisons between control and treatment groups were performed by the Mann–Whitney test, and a *p*-value < 0.05 was considered to be a statistical difference (*) between groups.

## Figures and Tables

**Figure 1 ijms-24-03400-f001:**
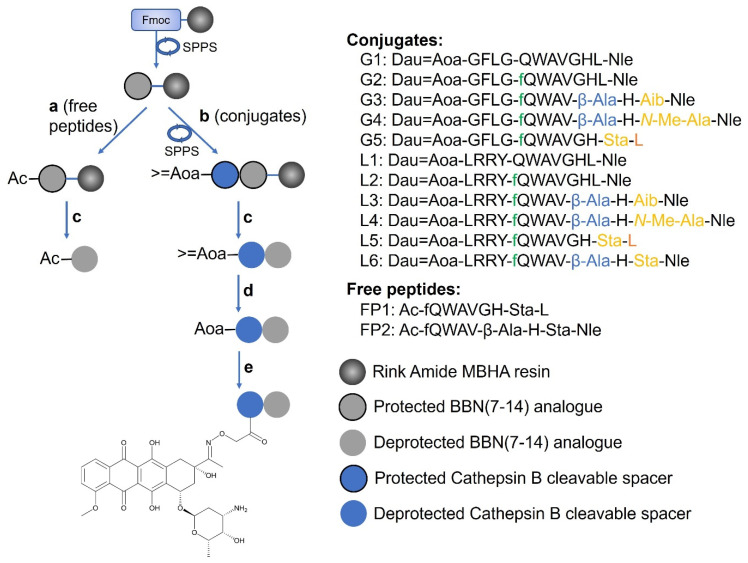
Synthesis of Dau-BBN (7-14) conjugates and BBN (7-14) analogues as free peptides. (**a**) Ac_2_O, DIPEA, DMF (1:1:3 *v*/*v*/*v*%), 30 min, RT. (**b**) (1) SPPS; (2) 3 eq >=Aoa-OH, 3 eq HOBt, 9 eq DIC in DMF, 1 h, RT. (**c**) TFA (95%), 2.5% TIS, 2.5% H_2_O, 2 h (**G1**–**G5**, **FP1**, **FP2**) or 10 mL TFA, 750 mg crystal phenol, 250 µL EDT, 500 µL thioanisole, 250 µL H_2_O (**L1**–**L6**). (**d**) Methoxyamine (1 M) in 0.1 M NH_4_OAc buffer (pH 5), 1-2 h, RT. (**e**) Dau (1.3 eq) in 0.1 M NH_4_OAc buffer (pH 5), o/n, RT. SPPS: Solid Phase Peptide Synthesis; Ac: acetyl; >=Aoa: isopropylidene aminooxyacetyl; Aoa: aminooxyacetyl; Aib: 2-aminoisobutyric acid; Sta: statine.

**Figure 2 ijms-24-03400-f002:**
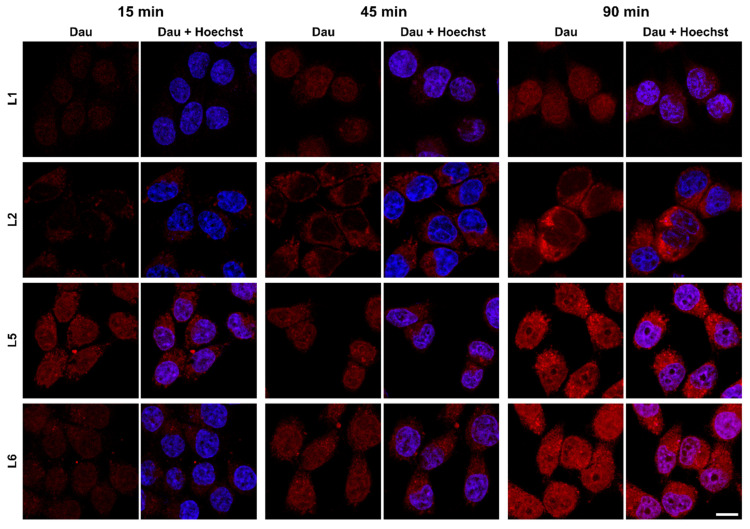
Localisation of Dau–BBN (7-14) conjugates in PC-3 cells, visualised by confocal laser scanning microscopy (CLSM). Conjugates were detected by the fluorescence of daunorubicin (red), nuclei were stained with Hoechst 33342 (blue). Scale bar: 10 μm.

**Figure 3 ijms-24-03400-f003:**
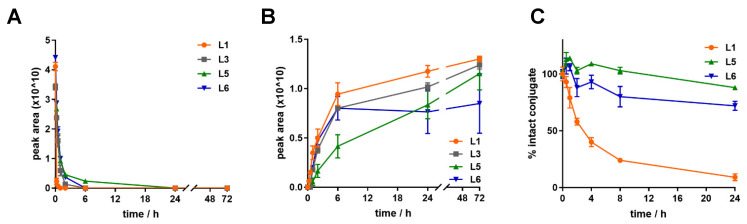
Stability profile of the conjugates in lysosomes and mouse plasma. (**A**) Degradation of **L1**, **L3**, **L5** and **L6** and (**B**) release of the active metabolite Dau=Aoa-Leu-OH in rat liver lysosomal homogenate (peak area, mean/10^10^ ± SD, n = 3). (**C**) Stability of compounds **L1**, **L5** and **L6** in mouse plasma over 24 h (%, mean ± SD, n = 3).

**Figure 4 ijms-24-03400-f004:**
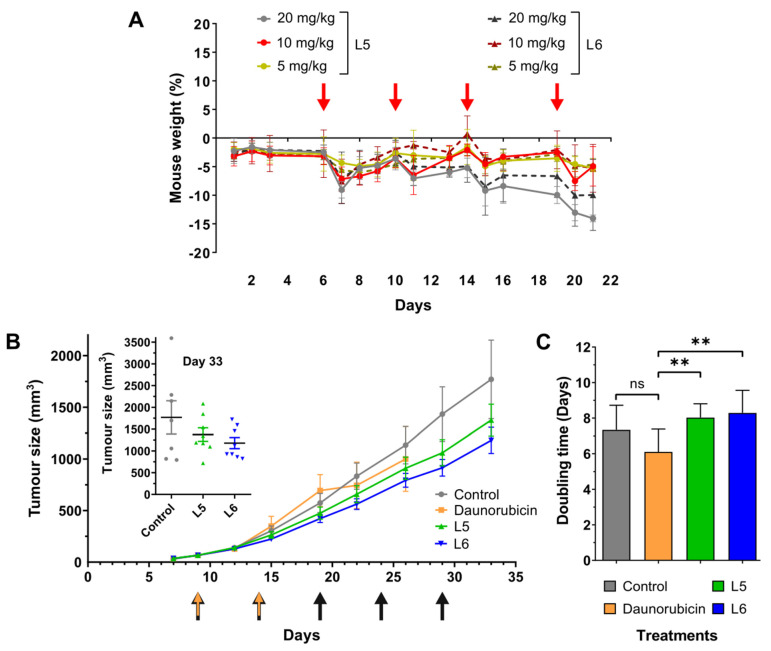
In vivo toxicity and antitumour effect of conjugates **L5** and **L6**. (**A**) Mouse body weight in chronic toxicity studies (%, mean ± SD, n = 3) after administration of either **L5** or **L6**, in three different concentrations, calculated on Dau content: 5 mg/kg, 10 mg/kg and 20 mg/kg. Four treatments (red arrows), three mice per group. (**B**) Tumour volume (mm^3^, mean ± SEM, n = 8). Administration regime: 0.9% saline (control group, black arrows), Dau (1 mg/kg, orange arrows) and PDCs (10 mg/kg calculated on Dau content, black arrows), every 5th day starting from day 9. (**C**) Calculated tumour doubling time for control, Dau- and PDC-treated groups (days, mean ± SD, n = 8). ns: non-significant difference. **: significant difference at *p* < 0.01.

**Table 1 ijms-24-03400-t001:** Chemical characterisation of LRRY (**L1**–**L6**) and GFLG (**G1**–**G5**) linker containing Dau-BBN (7-14) conjugates and free peptide analogues (**FP1** and **FP2**).

Code	(Dau=Aoa)-BBN(7-14) Compound/BBN(7-14) Compound	RP-HPLC Rt (min) ^a^	ESI-MS MW_meas_/MW_cal_ (g/mol) ^b^
G1	[Nle^14^]	24.35	1878.8988/1879.1050
G2	[D-Phe^6^, Nle^14^]	24.75	2025.9836/2026.2820
G3	[D-Phe^6^, β-Ala^11^, Aib^13^, Nle^14^]	23.45	2011.9672/2012.2550
G4	[D-Phe^6^, β-Ala^11^, *N*-Me-Ala^13^, Nle^14^]	23.42	2011.9494/2012.2550
G5	[D-Phe^6^, Sta^13^, Leu^14^]	24.25	2069.9928/2070.3350
L1	[Nle^14^]	20.93	2093.0528/2093.3780
L2	[D-Phe^6^, Nle^14^]	22.22	2240.1192/2240.5550
L3	[D-Phe^6^, β-Ala^11^, Aib^13^, Nle^14^]	21.23	2226.1036/2226.5280
L4	[D-Phe^6^, β-Ala^11^, *N*-Me-Ala^13^, Nle^14^]	20.58	2226.1018/2226.5280
L5	[D-Phe^6^, Sta^13^, Leu^14^]	22.35	2284.1462/2284.6080
L6	[D-Phe^6^, β-Ala^11^, Sta^13^, Nle^14^]	21.30	2298.1598/2298.6350
FP1	[D-Phe^6^, Sta^13^, Leu^14^]	21.76	1155.6368/1155.3690
FP2	[D-Phe^6^, β-Ala^11^, Sta^13^, Nle^14^]	21.17	1168.6534/1169.3960

^a^ Column: Phenomenex Aeris C18 column (250 mm × 4.6 mm) with 3.6 µm silica particle size (100 Å pore size); gradient: 0 min 2% B, 5 min 2% B, 30 min 90% B; eluents: 0.1% TFA in water (eluent A) and 0.1% TFA in acetonitrile–water (80:20, *v*/*v*) (eluent B); flow rate: 1 mL/min; detection at 220 nm. ^b^ Thermo Fisher Scientific Q ExactiveTM Focus hybrid quadrupole-orbitrap mass spectrometer.

**Table 2 ijms-24-03400-t002:** IC_50_ values related to the cytostatic effect of the Dau–BBN (7-14) conjugates and the free peptides on human breast cancer MDA-MB-231 and MDA-MB-453, and human prostate cancer PC-3 cell lines. All values are reported as mean ± SD (n = 2).

Conjugate/Free Peptide	MDA-MB-231 IC_50_ (µM)	MDA-MB-453 IC_50_ (µM)	PC-3 IC_50_ (µM)
G1	22.80 ± 3.12	11.45 ± 1.53	11.83 ± 2.50
G2	7.29 ± 3.41	12.72 ± 1.22	5.98 ± 2.10
G3	9.28 ± 0.13	8.78 ± 0.97	4.55 ± 0.76
G4	20.98 ± 0.55	7.37 ± 1.28	5.73 ± 0.24
G5	18.29 ± 1.46	11.62 ± 3.33	9.69 ± 0.17
L1	4.15 ± 0.18	7.87 ± 0.09	4.38 ± 0.33
L2	11.74 ± 0.09	19.14 ± 0.49	8.57 ± 1.61
L3	18.96 ± 3.23	>25	>25
L4	5.31 ± 0.01	21.21 ± 5.36	4.08 ± 0.09
L5	3.35 ± 0.32	5.86 ± 0.75	2.22 ± 0.19
L6	9.88 ± 2.82	9.64 ± 0.25	18.04 ± 3.01
FP1	>100	>100	>100
FP2	>100	>100	>100
Dau	0.90 ± 0.06	0.81 ± 0.07	0.75 ± 0.01

**Table 3 ijms-24-03400-t003:** UC_50_ values related to the uptake of the conjugates in the indicated human cancer cell lines after 1.5 h incubation. Each value indicates the concentration (μM) that corresponds to the internalisation of 50% of the conjugate inside the living cells.

Conjugate	MDA-MB-231 UC_50_ (µM)	MDA-MB-453 UC_50_ (µM)	PC-3 UC_50_ (µM)
G1	>25	>25	>25
G2	>25	17.04	10.50
G3	>25	22.59	>25
G4	23.33	20.22	>25
G5	>25	>25	>25
L1	18.67	11.63	22.92
L2	18.27	21.58	20.87
L3	>25	24.59	>25
L4	>25	>25	>25
L5	15.87	8.96	12.35
L6	15.47	4.15	16.09

## Data Availability

The data generated during this study are available by request to the corresponding author.

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
