# Peer review of "Targeting the Gastrin-Releasing Peptide Receptor (GRP-R) in Cancer Therapy: Development of Bombesin-Based Peptide–Drug Conjugates"

_ijms, 2023, doi:10.3390/ijms24043400_

Round 1

Reviewer 1 Report

The manuscript by Gomena et al., describes the design, development and biological testing of peptide-drug conjugates to target the GRP-Receptor in vitro and in vivo. Daunorubicin was elected as model drug; several conjugates were prepared, characterized and tested and two of them resulted promising in term of stability, cell internalization and cytostatic activity compared to the others.

The work is well written, the methodology and the results data are clearly presented.

Nevertheless, some concerns are raised by the limited cytostatic activity of the conjugates and the related IC50 values reported. In addition, the animal studies were performed with a drug amount 10 times higher than free drug (10 mg/kg versus 1 mg/kg). Since it looks that the conjugate is rapidly degraded into the lysosomal compartment, thus releasing the drug into the cytoplasm, how do the authors explain this finding? Is the poor cytotoxicity associated to lower cellular uptake of the conjugates than the free drug?

The use of a carrier should improve the transport and bioavailability of drugs with poor solubility/stability, and, at the same time, reduce the amount of drug to be administered thanks to more efficient and specific transport to the target site. Therefore, behind the impressive effort to design and build the conjugate, the biological data are less convincing about the real efficacy of the peptide-drug conjugate and the benefits of their use.

Please, the authors are invited to argument about these points.

Author Response

Thank you very much for the comments and suggestions. Please find, as follows, a point-by-point response to the questions.

Question 1: "[...], some concerns are raised by the limited cytostatic activity of the conjugates and the related IC50 values reported. In addition, the animal studies were performed with a drug amount 10 times higher than free drug (10 mg/kg versus 1 mg/kg). Since it looks like the conjugate is rapidly degraded into the lysosomal compartment, thus releasing the drug into the cytoplasm, how do the authors explain this finding? Is the poor cytotoxicity associated to lower cellular uptake of the conjugates than the free drug?"

Again, thank you for your comment, these are common questions related to PDCs. According to our observations, the free drug always has a lower IC50 value in an in vitro experiment when a drug like daunomycin can enter the cells by diffusion. The receptor levels are generally limited even if they are overexpressed, therefore, conjugates can enter the cells in a lower amount, as was demonstrated in several publications before [1] [2]. This was detected both by flow cytometry and confocal microscopy. In contrast, a conjugate might be more potent in comparison with the free drug when the free drug cannot enter the cancer cells efficiently or in the case of using multidrug resistant cells that pump out the free drug molecules. In addition, to obtain a comparable cytotoxic effect the free drug should be released from the conjugate. In the case of oxime-linked drug conjugates, there is no significant release of the free drug by enzymes in lysosomes, but an active metabolite will be released. (This fact also provides high stability of the conjugates in the circulation, so that we do not have to worry about an early drug release). This consists of the Dau connected to one amino acid through the aminooxyacetyl linker [3], in this case Dau=Aoa-Leu-OH. This metabolite binds to DNA, but not as efficiently as the free Dau. The saturation of the cellular uptake together with the lack of free drug release results in a lower cytostatic effect of the conjugate in contrast to the free drug. According to our observation, when the IC50 of a conjugate is not higher than one order of magnitude than in the case of Dau, they might have potential tumour growth inhibition effect in an in vivo experiment [4] [5]. It has to be underlined that the evaluation of the release of the active metabolite in vitro (lysosomal degradation assay), despite being a good indicator, might not directly correlate with the situation in vivo, since, for instance, we cannot know details about the level of expression of lysosomal enzymes or the lysosomal / endosomal escape of Dau=Aoa-Leu-OH in the xenograft models. It is also worth mentioning, that the oxime bond formation by chemoselective ligation is very effective and provides conjugates with excellent yields in higher amounts for in vivo studies, too. This construct with non-cleavable linker provides a suitable tool for comparison of the efficacy of the tumour homing peptides. After selection of the most potent targeting moiety, further effort can be done to increase the antitumour activity of the conjugates with other types of drug or linker systems. Dau is useful because of its autofluorescence properties and in this way the Dau-containing conjugate can be used for both cytostasis and cellular uptake studies.

It is right that we must use higher amount of conjugate (10 mg/kg Dau content in the conjugate) for in vivo studies. Usually, this is not the maximum tolerated dose for conjugates, while this value is only 1 mg / kg body weight for the free Dau. The in vivo experiments showed that the conjugates provided higher tumour growth inhibition than the free drug in maximum tolerated dose without significant toxicity: mice lost weight significantly during Dau treatment, therefore the experiment had to be finished earlier than in the case of mice treated with the conjugates. As we mentioned, this study is focused on the selection of the best homing peptides. Based on our results, further modifications can be done to improve the antitumour activity of the conjugates (e.g. changing drug molecule or linker system, optimization of the treatment schedule, etc.). In addition, we believe that in the case of targeted therapy similar to conventional chemotherapy, the combination of different conjugates might lead to a higher antitumour effect. Therefore, our best compounds can take a part in such types of mixtures.

Question 2: "The use of a carrier should improve the transport and bioavailability of drugs with poor solubility/stability, and, at the same time, reduce the amount of drug to be administered thanks to more efficient and specific transport to the target site. Therefore, behind the impressive effort to design and build the conjugate, the biological data are less convincing about the real efficacy of the peptide-drug conjugate and the benefits of their use."

As mentioned, daunomycin is a water-soluble drug that can easily diffuse through cell membranes, accessing the cytoplasm. On the other hand, our conjugates rely on receptor-mediated endocytosis to enter cancer cells. This process, due to the limited number of receptors (although overexpressed), seems to be less efficient compared to simple diffusion. Moreover, we demonstrated that, despite the higher content of Dau, our conjugates L5 and L6 do not cause significant toxicity in mice, while providing higher tumour growth inhibition than the free drug in its maximum tolerated dose. Therefore, we expect improved transportation of the payload to the tumour environment compared to healthy tissues, which would be beneficial in case of the attachment of more potent drugs to the homing peptide. In our next publications, it would be interesting to involve biodistribution studies to assess improved transportation of the drug to the tumour.

  1. Schuster, S., Biri-Kovács, B., Szeder, B., Buday, L., Gardi, J., Szabó, Z., Halmos, G., & Mező, G. Enhanced In Vitro Antitumor Activity of GnRH-III-Daunorubicin Bioconjugates Influenced by Sequence Modification. Pharmaceutics 2018, 10(4), 223.
  2. Schuster, S., Biri-Kovács, B., Szeder, B., Farkas, V., Buday, L., Szabó, Z., Halmos, G., & Mező, G. Synthesis and in vitro biochemical evaluation of oxime bond-linked daunorubicin-GnRH-III conjugates developed for targeted drug delivery. Beilstein J Org Chem. 2018, 14, 756–771.
  3. Orbán, E.; Mező, G.; Schlage, P.; Csík, G.; Kuli´c, Z.; Ansorge, P.; Fellinger, E.; Möller, H.M.; Manea, M. In vitro degradation and antitumor activity of oxime bond-linked daunomycin-GnRH-III bioconjugates and DNA-binding properties of daunomycin-amino acid metabolites. Amino Acids 2011, 41, 469–483.
  4. Ranđelović I, Schuster S, Kapuvári B, Fossati G, Steinkühler C, Mező G, Tóvári J. Improved In Vivo Anti-Tumor and Anti-Metastatic Effect of GnRH-III-Daunorubicin Analogs on Colorectal and Breast Carcinoma Bearing Mice. Int J Mol Sci. 2019 Sep 25;20(19):4763.
  5. Kiss K, Biri-Kovács B, Szabó R, Ranđelović I, Enyedi KN, Schlosser G, Orosz Á, Kapuvári B, Tóvári J, Mező G. Sequence modification of heptapeptide selected by phage display as homing device for HT-29 colon cancer cells to improve the anti-tumour activity of drug delivery systems. Eur J Med Chem. 2019 Aug 15;176:105-116.

Reviewer 2 Report

This study investigates the efficacy of bombesin analogs to target daunorubicin to cancer cell lines and in mice xenograft models. The work is logical, carefully planned, executed and documented and deserves publication with no substantial revisions. The therapeutic results are somewhat disappointing in terms of peptide drug conjugate potency in cells and in vivo. Determining binding affinities to GRP-R may explain this in addition to inefficient release of liberated drug following receptor uptake via cathepsin cleavage and endosomal escape. Continuation of this work with auristatin payloads and the self-immolative cathepsin linkers used in ADC would be interesting as the authors suggest.

Author Response

Thank you very much for the positive comments and precious suggestions.

The confocal microscopy images showed that the released Dau-containing metabolite can escape from lysosomes and accumulate in the nucleus. This localisation in time is conjugate-dependent and correlates with their cellular uptake, which might be influenced by the receptor binding affinity. Due to the improved stability of the oxime bond, there is no significant release of the free drug by enzymes in lysosomes, but an active metabolite will be released, consisting of Dau connected to one amino acid through the aminooxyacetyl moiety [1]. In our case, this would be Dau=Aoa-Leu-OH. Because the released metabolite was the same in all conjugates, the cathepsin cleavage and the endosomal / lysosomal escape should be the same and should not influence the biological activity. This is the reason why we use the same cathepsin B labile linkers to get results depending only on the homing peptide. In our next publication we will make more effort to focus on the study of receptor binding affinity as well.

Furthermore, the active metabolite Dau=Aoa-Leu-OH binds to DNA, but not as efficiently as the free Dau. This causes a shift of the IC50s of the conjugates to higher values compared to the free Dau. However, according to our observations, when the IC50 of a conjugate is within values that do not exceed one order of magnitude higher than that of Dau, it might have potential tumour growth inhibition effect in in vivo experiments [2] [3]. We showed that conjugates L5 and L6 at a 10 mg/kg Dau-content are non-toxic in mice and provide higher tumour growth inhibition than the free drug in its maximum tolerated dose (1 mg/kg). We believe that these results can also be improved in the case of the use of more potent payloads, such as auristatins, or combinations between conjugates containing drugs with a different mechanism of action.

Finally, establishing an appropriate preclinical model is crucial for translational cancer research due to the fact that the rate of successful clinical approval for cancer drugs is very low (approximately lower than 15%), especially in solid tumour [4]. The limitation of cell line-derived xenografts (CDX) model to not reflect the patient’s drug response sufficiently, would support the idea to use patient-derived xenografts (PDX) models to assess the efficiency of anti-tumor drugs in the specific microenvironment, since the models closely resemble the original tumours of patients [5]. However, this type of tumour bearing mice model is very expensive and rarely used for the selection of lead compounds in basic research. In our study we wanted to select a potent conjugate that might be efficient for prostate cancer. For this purpose, our experiments were suitable, and we hope that in the future we will be able to check our conjugates on such type of tumour bearing mice model.

  1. Orbán, E.; Mező, G.; Schlage, P.; Csík, G.; Kuli´c, Z.; Ansorge, P.; Fellinger, E.; Möller, H.M.; Manea, M. In vitro degradation and antitumor activity of oxime bond-linked daunomycin-GnRH-III bioconjugates and DNA-binding properties of daunomycin-amino acid metabolites. Amino Acids 2011, 41, 469–483.
  2. Ranđelović I, Schuster S, Kapuvári B, Fossati G, Steinkühler C, Mező G, Tóvári J. Improved In Vivo Anti-Tumor and Anti-Metastatic Effect of GnRH-III-Daunorubicin Analogs on Colorectal and Breast Carcinoma Bearing Mice. Int J Mol Sci. 2019 Sep 25;20(19):4763.
  3. Kiss K, Biri-Kovács B, Szabó R, Ranđelović I, Enyedi KN, Schlosser G, Orosz Á, Kapuvári B, Tóvári J, Mező G. Sequence modification of heptapeptide selected by phage display as homing device for HT-29 colon cancer cells to improve the anti-tumour activity of drug delivery systems. Eur J Med Chem. 2019 Aug 15;176:105-116.
  4. J A DiMasi, J M Reichert, L Feldman, A Malins. Clinical approval success rates for investigational cancer drugs. Clin Pharmacol Ther. 2013 Sep;94(3):329-35.
  5. Jaeyun Jung, Hyang Sook Seol, Suhwan Chang. The Generation and Application of Patient-Derived Xenograft Model for Cancer Research. Cancer Res Treat. 2018 Jan;50(1):1-10.